# Ultra-Narrow Metallic Nano-Trenches Realized by Wet Etching and Critical Point Drying

**DOI:** 10.3390/nano11030783

**Published:** 2021-03-19

**Authors:** Jeeyoon Jeong, Hyosim Yang, Seondo Park, Yun Daniel Park, Dai-Sik Kim

**Affiliations:** 1Department of Physics and Institute for Accelerator Science, Kangwon National University, 1 Gangwondaehak-gil, Chuncheon-si 24341, Gangwon-do, Korea; 2Department of Physics and Center for Atom Scale Electromagnetism, Ulsan National Institute of Science and Technology, Ulsan 44919, Korea; simi3070@snu.ac.kr; 3Department of Physics & Astronomy, Seoul National University, Seoul 08826, Korea; 4Department of Physics & Astronomy and Institute of Applied Physics, Seoul National University, Seoul 08826, Korea; remul17@gmail.com (S.P.); parkyd@snu.ac.kr (Y.D.P.)

**Keywords:** terahertz, nanoantennas, nano-trenches, critical point drying

## Abstract

A metallic nano-trench is a unique optical structure capable of ultrasensitive detection of molecules, active modulation as well as potential electrochemical applications. Recently, wet-etching the dielectrics of metal–insulator–metal structures has emerged as a reliable method of creating optically active metallic nano-trenches with a gap width of 10 nm or less, opening a new venue for studying the dynamics of nanoconfined molecules. Yet, the high surface tension of water in the process of drying leaves the nano-trenches vulnerable to collapsing, limiting the achievable width to no less than 5 nm. In this work, we overcome the technical limit and realize metallic nano-trenches with widths as small as 1.5 nm. The critical point drying technique significantly alleviates the stress applied to the gap in the drying process, keeping the ultra-narrow gap from collapsing. Terahertz spectroscopy of the trenches clearly reveals the signature of successful wet etching of the dielectrics without apparent damage to the gap. We expect that our work will enable various optical and electrochemical studies at a few-molecules-thick level.

## 1. Introduction

A pair of parallel metallic plates create a uniform electric field between them when a voltage difference is applied, provided that the separation between them is sufficiently small compared to the size of the plates themselves. They have not only become the central building block of electrochemical studies, but also greatly benefitted the field of nanophotonics, as the distribution of electromagnetic fields near a pair of parallel metallic plates can be expressed analytically and is exactly known. Therefore, metallic gap structures have provided an ideal testbed for quantitative analysis on light–matter interactions, as well as enhancing the interactions via strong electromagnetic field enhancements [1,2,3,4,5]. Especially, negative slot antennas on metallic films were widely employed in enhancing absorption of light [6,7,8,9], improving optical responses from semiconductors [10,11], exclusively probing surface dynamics separated from the bulk counterpart [12,13], and field enhancement-aided quantum plasmonics [14,15], etc. As the enhancement of light–matter interactions is inversely proportional to the gap width, much effort has been made to minimize the width of the slot antennas, ideally down to a ~1-nm level where a few-molecules-thick regime can be reached to reveal many exotic physical and chemical phenomena [16].

In such a context, the invention of atomic layer lithography made a huge breakthrough by making possible the experimental realization of wafer-scale arrays of horizontally aligned metal–dielectric–metal structures with a dielectric gap as small as 1 nm [17]. Selective wet etching of the dielectric layer afterwards can lead to the formation of ultra-narrow, empty metallic gaps with giant length-to-width and height-to-width aspect ratios, which we shall refer to as “metallic nano-trenches” hereafter [18]. These metallic nano-trenches have been demonstrated at gap widths as small as 5 nm and have been utilized as ultrasensitive detectors for liquid molecules and thermally active modulators for terahertz radiations [18,19]. Yet, for gap widths less than 5 nm, the method has not been very successful, mostly due to the catastrophic drying process after wet etching. The typical process for wet etching and drying is depicted in Figure 1a. A potassium hydroxide (KOH) solution is used to wet-etch alumina, which is the most used dielectric in the atomic layer lithography process. The solution does a great job in etching the alumina, but the strong capillary force exerted by the drying aqueous solution can interfere with the now free-hanging nano-trenches, leading to collapsing of the metallic plates. While well-controlled capillary collapsing may be utilized in fabricating plasmonic gaps down to 6-nm width [20], the process requires additional self-limiting geometry for the distance control and is better suited for point-like gaps, not so much for extended gap structures such as nano-trenches.

In this work, we demonstrate a means to overcome the issue and successfully fabricate metallic nano-trenches with gap widths as small as 1.5 nm. Here, we use the critical point drying (CPD) technique, whose working principle relies on replacing liquid water with liquid carbon dioxide (CO_2_) and vaporizing the CO_2_ at its critical point to minimize the force applied to a system in the drying process (Figure 1b). The CPD technique has been widely used in the field of biology and microelectromechanical systems (MEMS) for preventing capillary collapse of small structures including various microstructures [21,22,23,24] and freestanding thin films of semiconductors [25,26]. By applying this method to metallic nanostructures, we successfully fabricate 1.5-nm-wide metallic nano-trenches, which was unachievable with the conventional wet-etch and dry process.

## 2. Materials and Methods

Our metallic nano-trenches were fabricated with atomic layer lithography, which consisted of the following steps as described in Figure 2. Silicon wafers with high resistivity (>1000 Ohm·cm, MTI Korea, Seoul, Korea) were used as substrates because of the well-developed lithography processes and high transmission of long-wavelength radiation. We performed photolithography on the substrate to pattern an array of 10 × 40-μm-sized rectangles, separated from each other by 10 μm. Negative photoresist (image-reversed AZ5214E, Microchemicals, Ulm, Germany) was used to ensure smooth lift-off in the following steps. We then evaporated a 250-nm-thick gold film on the sample with electron beam evaporation (Korea Vacuum Tech., Gimpo, Korea). The subsequent lift-off process with acetone (Duksan Chemical, Incheon, Korea) led to a 250-nm-thick gold film with an array of 10 × 40-μm-sized rectangular holes separated by 10 μm from each other. The sidewall angle of the metallic layers should ideally be 90 degrees; in our sample, it was measured to be about 80 degrees, which was expected to cause a slight decrease in the transmitted amplitudes of the fabricated nano-trench structures [27]. Then, we conformally covered both the top and the sidewalls of the gold micropatterns with alumina using atomic layer deposition (ALD), with which we could control the thickness down to 1.5 nm. After that, a second layer of gold with 250-nm thickness was deposited on top of the whole structure. This filled the rectangular holes and defined a laterally aligned gold–alumina–gold structure at the sidewalls of the metallic layers. The top excess layer of gold deposited on top of the first gold pattern (i.e., outside the rectangular holes) was subsequently removed by applying an adhesive tape on top of the sample and peeling it. As the connection between the excess metallic layer and the rest of the sample was very weak, only the top excess metallic layer was removed, leaving the gold–alumina–gold structure intact. Next, the sample underwent argon ion milling at an oblique angle (80 degrees) to flatten the whole structure and minimize the roughness near the gap exit. This led to the formation of a horizontally aligned gold–alumina–gold gap structure, with the gap width solely determined by the thickness of the ALD-alumina. Note that the vertically aligned gap structure is beneficial for optical measurements, as an optical beam will be incident in a direction normal to the surface of the substrate. After the alumina-filled gap structure was formed, we wet-etched the sample by placing it in 1 M KOH (Duksan Chemical, Incheon, Korea) solution for 10~20 s. The sample was then transferred to a bath of deionized water (DI) to allow the reactants to diffuse out of the gap. After 24 h, the sample was transferred to a bath of isopropyl alcohol (IPA, Duksan Chemical, Incheon, Korea) to allow the gap to fill with IPA. Carefully transferring the sample to a CPD chamber and replacing the IPA with CO_2_ led to an empty metallic nano-trench structure. Note that the trenches had a <1:100 width-to-thickness ratio and a <1:6000 width-to-length ratio, which is nearly impossible to achieve with conventional fabrication tools such as focused ion beam milling or electron beam lithography.

The nano-trenches before and after wet etching of the dielectric were analyzed with terahertz (THz) time-domain spectroscopy. The spectroscopic method can non-destructively evaluate a gap structure without having to make an electronic contact and, therefore, prevents unwanted damage to the sample in the measuring process. In detail, we created THz pulses with 800-nm optical pulses from a Ti:sapphire oscillator and a DC-biased gallium arsenide photoconductive antenna. The THz pulses passed through a 1 × 1-mm rectangular aperture and the sample, subsequently, was incident on zinc telluride crystal for electro-optic detection. The detected field amplitude was transformed into a frequency domain spectrum via fast Fourier transform. All the measured spectra were normalized with respect to transmission spectra of the aperture plus bare silicon substrate, so that performance of the gap could solely be determined.

The final structure of our nano-trenches was an array of 1.5-nm-wide coaxial apertures with a rectangular perimeter of 10 × 40 μm, with separation of 10 μm from both sides, as depicted in Figure 3a. The polarization of the incident THz radiation was perpendicular to the long axis of the rectangle, which enabled efficient accumulation of charges at the gap via capacitive coupling. The coaxial apertures supported a fundamental TEM mode at a frequency determined by the perimeter of the aperture and the effective index inside the gap, which will be discussed later. Figure 3b,c show the scanning electron microscope (SEM, JEOL Ltd., Akishima, Japan) and transmission electron microscope (TEM, JEOL Ltd., Akishima, Japan) images of the sample before etching, respectively. While the roughness of the metallic layers was in the order of 10 nanometers or larger, the alumina layer deposited with the atomic layer deposition technique could completely cover the sidewalls of the metal, even at a thickness of 1 nanometer [17]. Therefore, the optical properties of the gap structure were still defined by the thickness of the alumina layer sandwiched between the two metallic layers. On the other hand, at the exit side of the gap, the small corrugations of the metallic layers may cause local collapsing of the trenches even with the smallest perturbation applied to the structure, which will then perturb the waveguide mode and lead to a strong modulation in the optical properties of the nano-trenches [28,29].

## 3. Results

The measured THz transmission spectra of the nano-trench samples before and after etching are summarized in Figure 4a. Before etching, the nano-trench sample shows a resonance peak at 0.4 THz with a transmitted amplitude of 0.3 (normalized with respect to the transmission of a bare silicon substrate). Note that the corresponding resonance wavelength of 750 μm is much longer than the perimeter of the nano-trenches (100 μm); this is due to an increased effective index of the gap caused by gap plasmon modes, as will be discussed later. Furthermore, following Kirchhoff’s integral formalism of diffraction, the measured transmission is directly related to near-field enhancement at the gap exit by the following equation [30]:FE = t/β.(1)
where FE denotes electric field enhancement at the gap, t is the transmitted amplitude measured at far field, and β is ratio of the gap area with respect to the total sample area. In our sample, β is in the order of 10^−4^, and therefore, the field enhancement at the gap reaches over 2000 at resonance, finding great potential in nonlinear optical studies or sensing applications.

When the gap undergoes the normal wet etching and blow dry process, we observe a significant decrease in transmission or equivalently decreased near-field enhancement in the nano-trenches (red line). As metallic nanogaps incorporate field enhancement via capacitive accumulation of charges [1], the decrease in field enhancement indicates that a conductive channel is formed between the two metallic layers, or that the gap has collapsed partially. Meanwhile, when CPD is used in the drying process, we see a completely different trend and observe an increase in the transmitted amplitude along with a blueshift of resonance (blue line). This indicates that the gap-filling dielectric has been removed successfully without any apparent damage to the nano-trenches, leading to a decrease in the permittivity inside the gap and a subsequent decrease in effective indices within the gap.

Figure 4b shows the theoretical spectra calculated with the coupled mode method [31,32], which have been successfully implemented in simulating nanostructures in long-wavelength regimes [33,34,35]. We modeled collapsing of the gap as an increase in imaginary permittivity of the gap material, i.e., an increase in conductivity, due to formation of metallic bridges in the gap. The results are in good agreement with the experimental data, supporting the validity of our analyses. It is worth noting that etching the alumina between the gap leads to an increase in amplitude as well, even though the absorption coefficient of alumina is negligible in THz frequencies. We attribute this to excitation of gap plasmon modes, which is a hybridization of two surface plasmon modes for the two closely placed metallic surfaces [36]. As the mode profile extends into the metal as well, ohmic losses from the metal may lead to an absorptive mode propagation even when the dielectric has no absorption [35]. Thus, both real and imaginary parts of effective indices decrease upon etching the dielectric, i.e., reduction in permittivity inside the gap, which is why we observed an increase in both the amplitude and frequency of the resonant mode of the nano-trenches upon the wet-etching (Figure 4c).

We also observed that the performance of the fabricated sample depends strongly on the etching time, i.e., the amount of time the sample stays in the wet etchant. This may be inferred from the fact that the alumina spacer exists not only between the metallic layers but also underneath one of the layers, which leads to a possible collapse of the nano-trenches upon complete etching of the underlying alumina layer (Figure 5a). Such a trend is observed in Figure 5b, where the transmission spectrum of an over-etched sample shows a drop in amplitude compared to a successfully etched sample, indicating formation of conducting channels like the case in Figure 4. The optimal etching time was found to be 10~20 s (Figure 5c); for etching times of 25 s or longer (not shown), both the resonance frequency and the transmitted amplitude decrease, indicating shrinkage and collapsing of the structure.

## 4. Discussion

The overall yield of our etching and CPD process is 90 percent for the 1.5-nm gap width and is limited by occasional damage to the sample caused by the turbulent flow of liquid CO_2_ inside the CPD chamber. Therefore, proper design of the sample holder and careful handling of the samples can further improve the yield of our process. Furthermore, as 1.5 nm is the smallest gap width achievable with our facilities, and since a smaller gap is much more prone to capillary collapses, we expect even higher success rates for nano-trenches with larger gap sizes. We briefly mentioned here that we also successfully applied our method on 5-nm-wide nano-trenches (data not shown), and the result was consistent with a previous report where 5-nm-wide nano-trenches were fabricated without the CPD process.

It is worth discussing the relatively short etching time of ~20 s for the nano-trench sample, considering that the gap width is only 1.5 nm wide and water or etchant molecules are expected to enter the gap very inefficiently. This can be understood in two aspects. First, the chemical reaction between KOH and alumina is extremely favored, which can be inferred from the reported etch rate of >2500 nm/min [37]. Therefore, the alumina in the gap is expected to attract the water molecules and ions into the gap. This situation is dramatically different from the case where the gap is empty and water molecules will be under a strong surface tension which blocks them from moving into the gap. Second, while we took the sample out of the KOH solution in ~20 s, the actual time that the gap-filling alumina reacts with KOH will be longer due to the finite diffusion coefficient of hydroxide ions in water. Therefore, the etching process is expected to continue to some extent even after the sample has been transferred to the DI bath. This implies that there might be better choices of etchant and etching time for higher throughput and smaller achievable widths, which is a topic of our future study.

Our scheme could also be improved by further optimization of the fabrication parameters. For instance, using a negative photoresist instead of image-reverse resist will simplify the procedure and may lead to better control over the geometry of the metallic layers. Furthermore, one could use electroplating instead of lift-off to achieve a steeper sidewall angle of the metallic layer, which can improve the optical properties and mechanical stability of the fabricated nano-trench structure. Such improvements may lead to realization of nano-trenches with even smaller gap widths and stronger field enhancement, which could open a new venue towards strong interaction of terahertz waves with a single-molecule-thick layer.

In summary, we developed a method for fabricating high-aspect-ratio metallic gaps, or nano-trenches, as narrow as 1.5 nm. This was achieved by fabrication of a metal–dielectric–metal gap structure and subsequent wet etching of the gap-filling dielectric. We found that the CPD technique efficiently eliminates the surface tension of water in the drying process, which, if present, leads to collapsing of the nanostructure. Successful etching or collapsing of the nano-trenches could be non-destructively determined with THz spectroscopy, as the optical properties of the gap structure are very sensitive to the formation of conducting channels. Unlike other spacer-based metallic gap structures, the nano-trenches have no filling material within the gap, which means that the optical hot spot is freely accessible to various types of materials. Therefore, the nano-trenches could find numerous potential applications in ultrasensitive molecular detection, nonlinear optics, and electrochemical studies.

## Figures and Tables

**Figure 1 nanomaterials-11-00783-f001:**
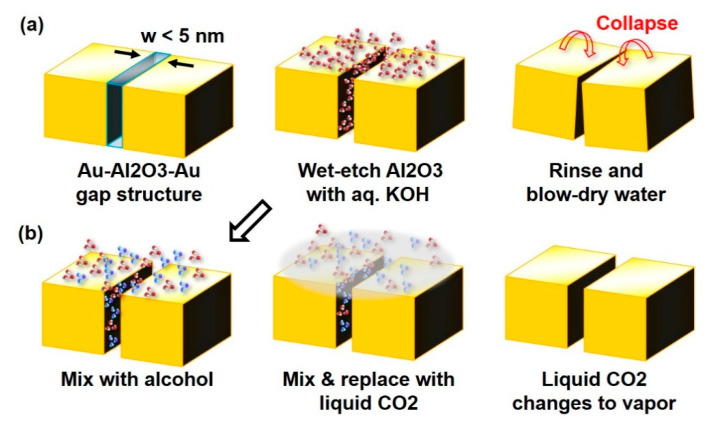
Schematic of the wet-etching process for metal–dielectric–metal gap structure, (**a**) without and (**b**) with the critical point drying (CPD) process. Due to the surface tension of the water, the gap collapses during the drying process when the gap is smaller than 5~10 nm. Using the CPD technique replaces the gap-filling material with liquid CO_2_ and removes the surface tension, thereby enabling metallic nano-trenches with a gap width as small as 1.5 nm.

**Figure 2 nanomaterials-11-00783-f002:**
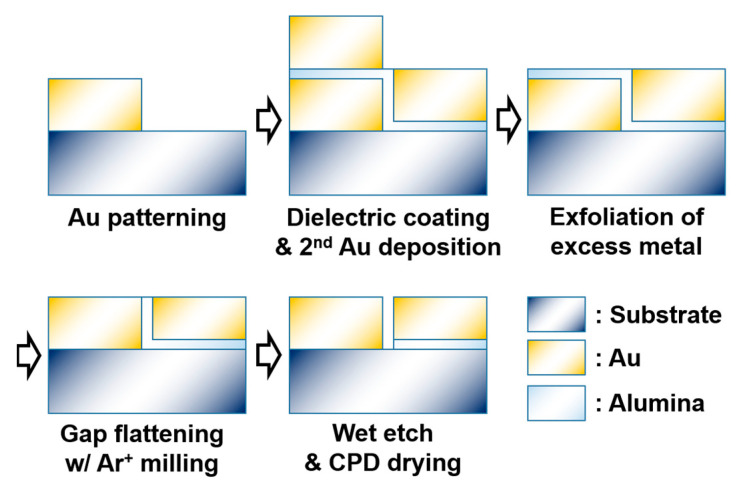
Fabrication scheme of the nano-trenches in a cross-sectional view.

**Figure 3 nanomaterials-11-00783-f003:**
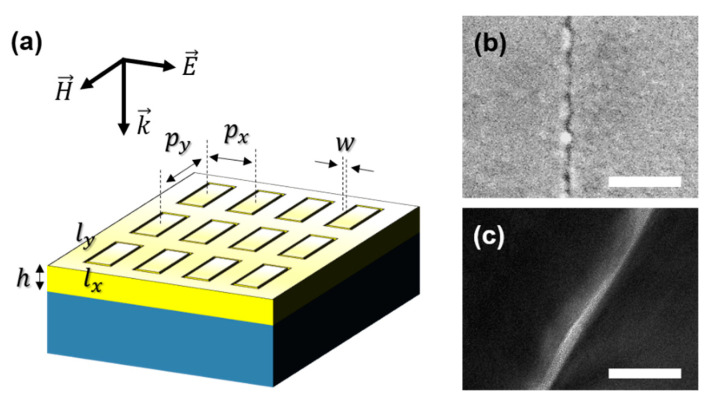
(**a**) Overview of the fabricated nano-trench array on Au film on a silicon substrate. In our sample, w = 1.5 nm, px = 20 μm, py = 50 μm, lx = 10 μm, ly = 40 μm, h = 250 nm. Polarization is perpendicular to the long side of the rectangular ring (ly). (**b**,**c**) Top-view SEM and cross-sectional TEM images of the metal–insulator–metal gap structure, respectively. Scale bar: (**b**) 50 nm; (**c**) 20 nm.

**Figure 4 nanomaterials-11-00783-f004:**
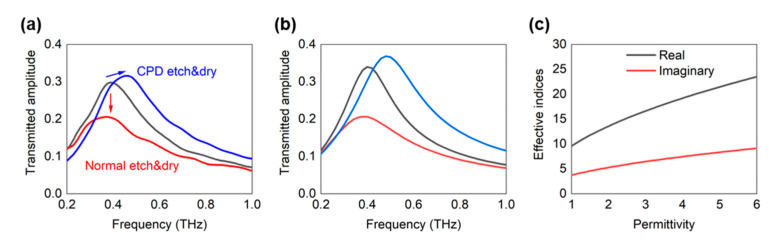
(**a**) Measured terahertz (THz) transmission spectra of 1.5-nm-wide nano-trenches before etching (black line), after wet etch and blow dry (red line), and after wet etch and CPD drying processes (blue line). Reduced transmission with no change in resonance frequency indicates damages in the gap, while increased transmission with a blueshift in the resonance shows successful removal of the dielectric between the two metallic layers. (**b**) Corresponding theoretical spectra calculated with coupled mode method. (**c**) Effective mode indices within a 1.5-nm-wide gap as a function of permittivity of the gap-filling material. The real (imaginary) part of the effective index induces frequency shift (amplitude change) of the resonance peak.

**Figure 5 nanomaterials-11-00783-f005:**
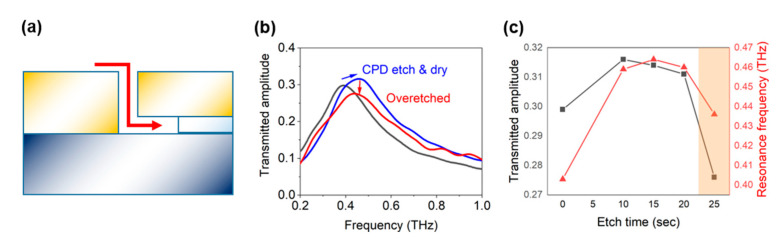
(**a**) Schematic of an over-etched sample. As there is a layer of alumina not only between but also beneath the metallic layers, etching for a longer time leads to undermining of the whole structure and partial collapsing of the nano-trenches. Note that the figure is not in scale and the vertically aligned portion of the dielectric (250 nm) is much smaller than that of the underlying layer (10 μm). (**b**) Spectra of the nano-trenches fabricated with proper wet etching time (blue line) and longer etching time (red line). (**c**) Transmitted amplitude and frequency at the resonance of the nano-trenches, fabricated with different etching times. Wet etching time of longer than 25 s leads to partial collapsing of the structure (shaded area).

## Data Availability

The data presented in this study are available from the corresponding author.

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
