# Peer review of "Ultra-Narrow Metallic Nano-Trenches Realized by Wet Etching and Critical Point Drying"

_nanomaterials, 2021, doi:10.3390/nano11030783_

Round 1

Reviewer 1 Report

I find the manuscripti interesting but the experimental results are rather thin and do not render enough support for the claims.

Preventing capillary collapse of small structures has been studied in the MEMS community for almost three decades, and critical point drying has been one of the solutions all along, see e.g.

Supercritical Carbon Dioxide Drying of Microstructures

By: Mulhern, G. T.; Soane, D. S.; Howe, R. T.

Conference: Proc. 7th International Conference on Solid-State Sensors and Actuators

Pages: ‏ 296-299   Published: ‏ 1993

Mechanical stability and adhesion of microstructures under capillary forces. I. Basic theory

By: Mastrangelo, C.H.; Hsu, C.H.

Journal of Microelectromechanical Systems ‏  Volume: ‏ 2   Issue: ‏ 1   Pages: ‏33-43   Published: ‏ March 1993, Times Cited: 366

Mechanical stability and adhesion of microstructures under capillary forces. II. Experiments

By: Mastrangelo, C.H.; Hsu, C.H.

Journal of Microelectromechanical Systems ‏  Volume: ‏ 2   Issue: ‏ 1   Pages: ‏ 44-55   Published: ‏ March 1993

Resist collapse is similarly an old research topic, see e.g.

Aqueous-based photoresist drying using supercritical carbon dioxide to prevent pattern collapse

By:Goldfarb, DL (Goldfarb, DL); de Pablo, JJ (de Pablo, JJ); Nealey, PF (Nealey, PF); Simons, JP (Simons, JP); Moreau, WM (Moreau, WM); Angelopoulos, M (Angelopoulos, M)

JOURNAL OF VACUUM SCIENCE & TECHNOLOGY B, Volume: 18, Issue: 6, Pages: 3313-3317, 2000

DOI: 10.1116/1.1313582

Abstract

A supercritical drying process was developed to eliminate the capillary forces naturally present during normal drying of photoresist materials. Supercritical carbon dioxide (scCO(2)), organic solvents and surfactants were used to prevent the collapse of high-aspect-ratio structures fabricated from aqueous-based photoresist. Nondistorted resist lines were patterned with this process with aspect ratios of at least 6.8. Water rinsed resist structures cannot be dried directly with scCO(2) due to the low solubility of water in the supercritical phase.

Capillary collapse, on the other hand has been utilized in making small-gap nanoplasmonic devices, see e.g.

Kim et al. Capillary-force-induced collapse lithography for controlled plasmonic nanogap structures,

Microsystems & Nanoengineering ( 2020) 6:65 Microsystems & Nanoengineering

https://doi.org/10.1038/s41378-020-0177-8 www.nature.com/micronano

So the background and history of the story have to be rewritten to reflect the rich history of the topic.

On the more technical side, I have a few observations:

“First, we pattern an array of 10 um x 40 um-sized rectangular holes separated by 10 um from each other on a 250 nm-thick gold film by photolithography and lift-off process on a silicon substrate.”

The accuracy of this description is not enough.

One possible process flow deduced from the above is this:

250 nm thick gold film deposition

optical lithography

gold evaporation

resist lift-off

But I guess the process flow is:

start with a silicon wafer

perform lithography

evaporate gold

do lift-off

No details are given, even though they are of paramount importance.

What resist ? What thickness ? What lithography ? What resist sidewall angle ? What resist sidewall roughness ?

Why not gold electroplating ?  Then you could optimize for straight vertical resist sidewalls. With lift-off, resist sidewall angle is typically tapered (negative slope) and this will make top view optical experiments more difficult.

It seems from SEM and TEM figures that sidewall roughness is in fact much larger than 1.5 nm, and the gap is determined by resist roughness and not by ALD film thickness. Or rather, a tortuous path defined by resist roughness is truly copied by the ALD film.

Why is there no study as a function of linewidth ? If you claim a new patterning process, demonstrating one single structure with one single linewidth is hardly enough.

Reviewer 2 Report

In this article, authors introduce a novel method to create metallic nano-trenches with widths as small as 1.5 nm, by a critical point drying technique. I suggest to publish the paper after authors address the following questions.

  • How is the yielding rate to create the 1.5nm trench?
  • What is the thickness of the 2nd gold layer?
  • How did authors remove the top excess layer of gold by the mechanical exfoliation? Is it a chemical mechanical polishing process? Authors should describe the process clearly.
  • How did authors ensure that IPA has filled with the trench after 24 h of soaking?
  • What is uniformity of process? How large is the 1.5nm trench area that author can create?
  • How is the repeatability of the process? Why there is standard deviation for Fig. 5c. How many test did authors perform?

Round 2

Reviewer 1 Report

I am happy with the revised version.